# Droplet Spreading Characteristics on Ultra-Slippery Solid Hydrophilic Surfaces with Ultra-Low Contact Angle Hysteresis

**Yajie Song [1], Qi Wang [1], Yushan Ying [1], Zhuo You [2], Songbai Wang [1], Jiang Chun [1], Xuehu Ma [1] and Rongfu Wen [1,*]**

[1] Liaoning Key Laboratory of Clean Utilization of Chemical Resources, Dalian University of Technology, Dalian 116024, China; songyajielady@mail.dlut.edu.cn (Y.S.); 15524572525@126.com (Q.W.); 563674189@mail.dlut.edu.cn (Y.Y.); 32045115@mail.dlut.edu.cn (S.W.); chunjiang@mail.dlut.edu.cn (J.C.); xuehuma@dlut.edu.cn (X.M.)

[2] Wuhan Second Ship Design and Research Institute, Wuhan 430205, China; zhuoyouzju@zju.edu.cn

[*] Correspondence: rongfuwen@dlut.edu.cn

**Abstract:** Dynamic interactions of the droplet impact on a solid surface are essential to many emerging applications, such as electronics cooling, ink-jet printing, water harvesting/collection, antifrosting/icing, and microfluidic and biomedical device applications. Despite extensive studies on the kinematic features of the droplet impact on a surface over the last two decades, the spreading characteristics of the droplet impact on a solid hydrophilic surface with ultra-low contact angle hysteresis are unclear. This paper clarifies the specific role of the contact angle and contact angle hysteresis at each stage of the droplet impact and spreading process. The spreading characteristics of the droplet impact on an ultra-slippery hydrophilic solid surface are systematically compared with those on plain hydrophilic, hydroxylated hydrophilic, and plain hydrophobic surfaces. The results reveal that the maximum spreading factor ($\beta_{max}$) of impacting droplets is mainly dependent on the contact angle and *We*. $\beta_{max}$ increases with the increase in *We* and the decrease in the contact angle. Low contact angle hysteresis can decrease the time required to reach the maximum spreading diameter and the time interval during which the maximum spreading diameter is maintained when the contact angles are similar. Moreover, the effect of the surface inclination angle on the spreading and slipping dynamics of impacting droplets is investigated. With the increase in the inclination angle and *We*, the gliding distance of the impacting droplet becomes longer. Ultra-low contact angle hysteresis enables an impacting droplet to slip continuously on the ultra-slippery hydrophilic surface without being pinned to the surface. The findings of this work not only show the important role of the surface wettability in droplet spreading characteristics but also present a pathway to controlling the dynamic interactions of impacting droplets with ultra-slippery hydrophilic surfaces.

**Keywords:** droplet impact; spreading; dynamic interactions; contact line pinning; ultra-slippery surface; low contact angle hysteresis; maximum spreading factor; surface wettability

## 1. Introduction

Dynamic interactions and spreading characteristics of impacting droplets on a solid surface are ubiquitous phenomena in nature and have attracted attention due to their broad-reaching traditional and emerging industrial applications, such as spray cooling for the thermal management of electronics devices, semiconductor chips, lasers, and turbine blades, ink-jet printing and 3D printing, fuel-injection and annealing, heat transfer enhancement and chemical reactions, water harvesting and desalination, microfluidic systems, and biomedical devices [1–8]. Microfabrication of advanced materials, solder bumps on printed circuit boards, electric circuits in microelectronics, and ice accumulating on power lines also involve interactions between impacting droplets and the solid substrate. Typical applications can also be found in evaporators of the multi-effect distillation system for seawater desalination and trickle bed reactors of the chemical catalytic reactions for material

fabrication. Understanding the physical phenomena and mechanisms of a liquid spreading on a solid surface is of utmost importance to improving the efficiency of processes and developing advanced technology for various industrial systems [9,10].

Worthington [11,12] was one of the first to conduct systematic experiments for observing the phenomenon of a droplet impacting upon a solid surface. Since then, driven by the interest in the droplet impact phenomenon and the importance of droplet spreading processes in numerous industrial applications, research on the droplet impact and corresponding dynamic behaviors on the substrate has been conducted [13–17]. When a droplet impacts upon a solid surface with a moderate velocity, it undergoes an initial deformation that spreads up to the maximum contact area and then relaxes to an equilibrium shape or partially/totally rebounds, depending on the wetting properties. The outcome of a droplet impacting upon a solid surface is dependent on the velocity and direction of the impacting droplet relative to the surface, the physical properties and size of the impacting droplet, and the physical and chemical properties of the solid surface [18–20]. As one of the key parameters that describe the droplet impact process, the Weber number is a dimensionless quantity that compares the kinetic energy and surface energy of a droplet impacting upon the substrate. The maximum spreading factor of the impacting droplet is another important indicator that can be used to directly describe the liquid spreading capability.

The effect of surface properties on liquid spreading characteristics has been widely studied in order to improve the spreading capability of the droplet impact [21–23]. Antonini et al. [24] experimentally studied the effect of the surface wettability and Weber number on the spreading behaviors of the droplet impact. The results indicate that the surface wettability determines both the maximum spreading factor and the spreading time of the droplet impact with a *We* number between 30 and 200. A higher Weber number can weaken the effect of the surface wettability due to the increase in the inertial force. The role of the surface capability in droplet spreading has also been investigated in related studies with numerical simulations, e.g., the VOF method and the lattice Boltzmann (LB) method [25–27]. Liang et al. [27] simulated a droplet impacting upon surfaces with different surface wettabilities and the results show that droplet rebound could easily be observed on non-wetting surfaces. A pseudopotential model of the LB method used by Quan et al. [28] also shows that high hydrophobicity leads to an enhancement of the droplet retraction speed and rebound. Du et al. [29] studied the effect of the impact velocity, fluid viscosity, and material wettability on the spreading and retraction dynamics using COMSOL Multiphysics® software. The results show that the retraction rate is strongly affected by the material wettability. Another main conclusion of the effect of the surface wettability on the droplet impact is that an increase in the contact angle from hydrophilic to hydrophobic has a considerable effect on the liquid film's geometry and the lamella's formation. Zhang et al. [30] found that the surface wettability contributes to the elevation of the lamella and has a significant effect on the spreading and splashing properties. For $\theta_a < 90°$, the splashing threshold $K = OhRe^{1.25}$ is independent of the surface wettability; but for $\theta_a > 90°$, $K$ is proportional to $\cos\theta_a$. Lin et al. [31] investigated the effect of liquid viscosity and surface wettability on the droplet spreading factor, spreading time, and oscillation. Wang et al. [32] studied the retraction dynamics of water droplets and proposed two different retraction modes: inertial mode (a rim–lamella structure) and capillary mode (a collapsed rim and lamella and capillary wave propagation). The retraction process of hydrophilic surfaces is dominated by capillary modes at low *We* numbers and inertial modes at high *We* numbers. For both hydrophobic and superhydrophobic surfaces, the capillary contraction mode always dominates the droplet retraction process. In addition, experimental and theoretical studies of a droplet impacting upon a spherical substrate have been conducted [33].

Recent advancements in micro/nanomaterials and precision manufacturing have enabled the judicious construction of various interfacial materials with the desired structural features and composition in order to modify the physical and chemical properties of the substrate [34–39]. The exciting progress in the development of functional coatings and surfaces has directly prompted the emergence of many intriguing dynamic phenomena

relevant to liquid and solid wetting states that are different from those on traditional solid surfaces [40,41]. As one of the extreme wetting states, superhydrophobicity, also known as the lotus effect, can greatly reduce the spreading area of the droplet impact and help the droplet to bounce off the solid surface owing to the presence of the trapped air layer between the impacting droplet and the underlying substrate [42,43]. The impact process of droplets on superhydrophobic surfaces usually manifests in two stages: spreading and rebounding. Scholars' research on the dynamics of the droplet impact on superhydrophobic surfaces mainly focuses on the deformation, rebound height, and contact time during the droplet impact process. In contrast, superhydrophilic surfaces can make liquid water spread rapidly to cover the solid substrate as much as possible, which is of importance in various practical applications [44,45]. A recent study by Chun et al. [39] proposed a superhydrophilic surface with nanowire bundles and a V-groove to make the droplets spread rapidly, resulting in a much larger spreading area. Wang, et al. [46] fabricated a three-dimensional ZnO hierarchical nanopillar structure, resulting in a 3-fold improvement in the wicking properties. The capillary flow will be obstructed when the nanowires on adjacent nanopillars are long enough to cross. Zheng et al. [47] combined the action of the micropillar capillary driving force and penetration inside the nanopores to enhance the spreading of droplets by constructing nanopore structures on micropillars. However, nanopores with a greater depth can lead to higher permeability and a slower spreading speed. For such a complete spreading of liquid on the solid surface, it is generally accepted that there is a precursor liquid film propagating in front of the spreading liquid. However, the local pinning effect induced by micro/nanostructures or defects on the solid surface can hinder the formation of the precursor film, and, consequently, limit the infinite spreading of the liquid [48,49].

In contrast to the concept of enhancing the spreading of a liquid on a superhydrophilic surface by constructing micro/nanostructures, the local contact line pinning of the liquid film/droplet on a solid surface can also be reduced by decreasing the contact angle hysteresis using lubricant-infused slippery surfaces [50–52]. Guo and Zhang et al. [53,54] demonstrated the rapid removal of a droplet on a hydrophilic surface modified by a hydrophilic liquid lubricant. The results show that the spontaneous movement of highly wetted liquids with low contact angle hysteresis can be obtained using flexible polymers with a gradient grafting density. In addition to the coupling of micro/nanostructures and an entrapped liquid lubricant, a hydrophilic solid surface with low contact angle hysteresis can also be achieved simultaneously by constructing surfaces with a high degree of chemical uniformity (at the molecular level) and a high degree of physical uniformity (at the structural level) [55–58]. Ho et al. [55] found that water molecules can be made to migrate on the surface with a small energy barrier by constructing uniformly distributed high-density adsorption sites on the hydrophilic surface, thus reducing the movement resistance of contact lines on the surface. It was demonstrated that the low-hysteresis slippage of liquid can be realized on the hydrophilic surface. By improving the iCVD method for grafted polymer coatings, Khalil et al. [56] minimized the contact angle hysteresis of low-surface-tension droplets, improving the condensation heat transfer performance. Cha et al. [57] grafted PEG silane onto a plain silicon surface, achieving hydrophilicity and low contact angle hysteresis at the same time. Kaneko et al. [58,59] prepared a hybrid film with static hydrophilicity but low contact angle hysteresis with a sol–gel solution of PEG10 silane and TEOS. The results show that the longest PEGn-Si chain ($n = 9$–12) had the smallest contact angle hysteresis. Previous research has demonstrated that ultra-slippery hydrophilic solid surfaces with low contact angle hysteresis exhibit excellent heat/mass transfer performance in condensation and defrosting. However, the underlying mechanism of the spreading dynamics of droplets on ultra-slippery hydrophilic solid surfaces remains poorly understood. Despite extensive studies on the kinematic features of the droplet impact over the last two decades, research on the effect of the surface wettability on the droplet impact dynamics mainly focuses on the role of the advancing contact angle in the deformation, maximum spreading diameter, and droplet retraction dynamics during the

droplet impact process. The role of low contact angle hysteresis as the main parameter of the surface wettability during the droplet impact process remains unclear. A better understanding of the role of the contact angle and contact angle hysteresis in the droplet spreading characteristics is vital for the design of advanced functional coatings and surfaces for various emerging applications.

In this work, an ultra-slippery hydrophilic surface with low contact angle hysteresis was fabricated to achieve the rapid spreading of droplets without contact line pinning on the solid substrate. The spreading characteristics of the droplet impact on the ultra-slippery hydrophilic surface are systematically compared with those on plain hydrophilic, hydroxylated hydrophilic, and plain hydrophobic surfaces through high-speed visualization experiments. The effects of the *We* number, contact angle, and contact angle hysteresis on the maximum spreading factor and the time interval during which the maximum spreading diameter of the droplet impacting upon the surface is maintained were identified and are discussed. In addition, the effect of the surface inclination angle and Weber number on the spreading and slipping behaviors of impacting droplets was investigated. Understanding the effect of contact angle hysteresis on the droplet slipping behavior may provide insights that help us explore advanced enhancement strategies for the spreading and transportation of liquids.

## 2. Surface Fabrication and Methods

### 2.1. Materials and Surface Fabrication

Figure 1a shows the preparation process of the four surfaces measured in this work. Original silicon wafers (P type, $\langle 100 \rangle$ orientation, 0 to 20 ohm·cm, $650 \pm 10$ µm thick, single-side-polished, Zhejiang Lijing Photoelectric Technology Co., Ltd., Hangzhou, China) taken directly from the package with a native oxide layer were cut into 2 cm by 2 cm pieces, cleaned in acetone, ethanol (>99.7%, Tianjin Kemiou Chemical Reagent Co., Ltd., Tianjin, China), and deionized water for 5 min in an ultrasonic bath, and dried with nitrogen to obtain the plain hydrophilic (plain-HI) surfaces. The cleaned silicon wafers were treated with oxygen plasma (20 W) for 10 s and then placed in air for one day to obtain the hydroxylated hydrophilic (hydroxyl-HI) surfaces. The dried silicon wafers were cleaned with oxygen plasma (300 W) for 60 s. The plasma-treated silicon wafers were immersed in a solution of 1 µL of 2-[methoxy (polyethyleneoxy)$_{9-12}$propyl] trimethoxysilane (Gelest Inc., Morrisville, PA, USA), 8 µL of hydrochloric acid (36.0~38.0%, Beijing Chemical Works, Beijing, China), and 10 mL of toluene (>99.0%, Beijing Chemical Works, Beijing, China) for 18 h at room temperature. By a "grafting to" method [57,60,61], the hydroxylated surfaces were covalently bonded to the PEG silane via O–Si bonds to form PEG brushes on the substrate. After the reaction, the silicon wafers were repeatedly rinsed to obtain the ultra-slippery hydrophilic (ultra-SHI) surfaces, and then the surfaces were soaked in deionized water to preserve them. As a comparison, a plasma-treated silicon wafer (cleaned with oxygen plasma (300 W) for 60 s) and 100 µL of trichloro-(1H,1H,2H,2H-perfluorooctyl)silane (97.0%, Alfa Aesar (China) Chemicals Co., Ltd., Shanghai, China) were placed into a vacuum dryer at a vacuum pressure of 0.8 kg/cm$^2$ for 2 h to obtain plain hydrophobic (plain-HO) surfaces by chemical vapor deposition. The average roughness values, Ra and Rq, of the plain-HI surfaces, plain-HO surfaces, and ultra-SHI surfaces were Ra = 0.6 nm and Rq = 0.7 nm, Ra = 0.5 nm and Rq = 0.6 nm, and Ra = 0.4 nm and Rq = 0.5 nm, respectively. As shown in Figure 1d, the silicon surfaces grafted with PEG silane had the lowest nanoscale surface roughness when compared with the other test surfaces, showing a smooth morphology and no aggregation or defects on the surface. Of course, the fabrication method for ultra-slippery hydrophilic (ultra-SHI) surfaces used in this work can only be carried out on silicon wafers because not all surfaces retain the plasma activation and the substrate cannot be damaged by the acid or toluene in the solution. In addition, the PEG silane is easily washed away because it is grafted to the substrate by chemical bonds. The low contact angle hysteresis of the ultra-SHI surface is demonstrated only for water as a working fluid.

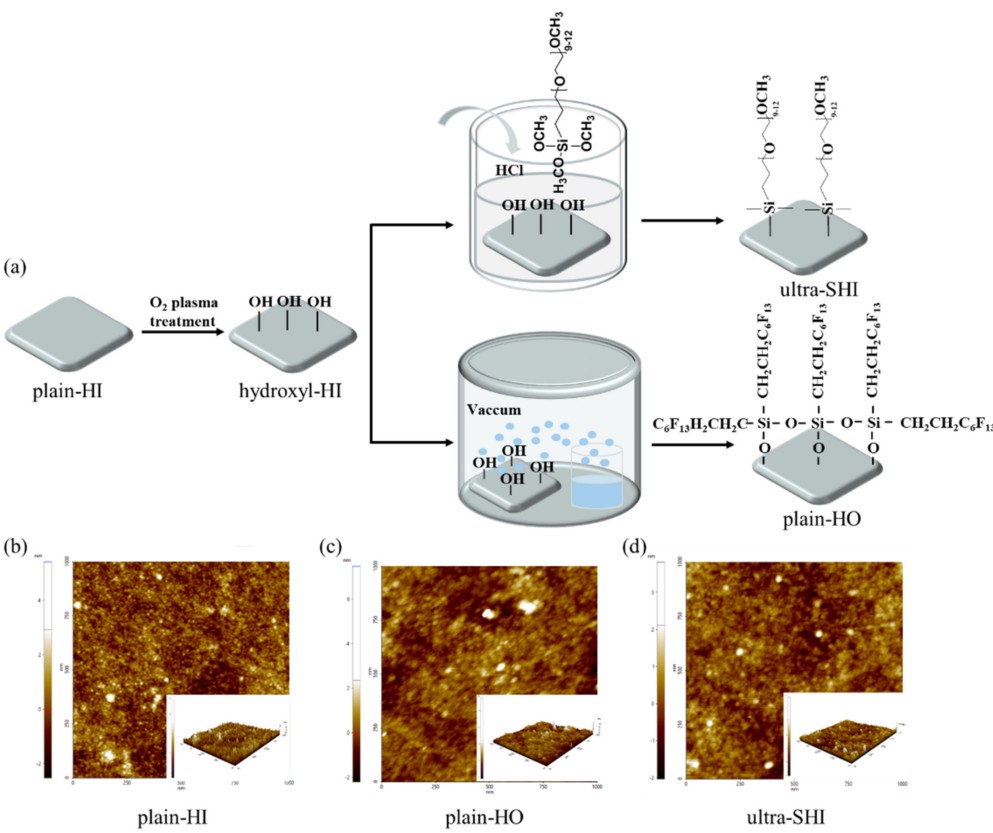

**Figure 1.** Fabrication and characterization of test surfaces. (**a**) Fabrication processes for the plain hydrophilic (plain-HI) surface, the hydroxylated hydrophilic (hydroxyl-HI) surface, the plain hydrophobic (plain-HO) surface, and the ultra-slippery hydrophilic (ultra-SHI) surface. AFM images showing the surface roughness of the (**b**) plain-HI surface, (**c**) plain-HO surface, and (**d**) ultra−SHI surface.

Table 1 summarizes the modification methods for silicon surfaces and water contact angle data reported in the literature. The reaction mechanism is that the hydrolyzable groups (such as $-OCH_3$) in the silanized polymer chain are hydrolyzed by the acid in the solution to produce the alcohol groups. The surface silanol groups and the end alcohol groups are covalently bound, grafting the polymer onto the surface. The properties of the polymer layer (e.g., the water contact angle) can be changed due to differences in the polymers and reaction conditions (such as the polymerization time and monomer concentration).

**Table 1.** Surface modification methods reported in the literature.

| Substrate | Pretreatment | Reaction Conditions | $\theta_a/\theta_r$ (°) Water | Ref. |
|---|---|---|---|---|
| Silicon wafers | Concentrated sulfuric acid containing sodium dichromate and hydrogen peroxide | Grafting: $(CH_3)_3SiCl$, toluene, room temperature, 72 h | 105/96 | [62] |
| | | Grafting: $(CH_3)_3SiN(CH_3)_2$, toluene, room temperature, 72 h | 106/98 | |
| | | Grafting: $(CH_3)_3SiOSO_2CF_3$, toluene, room temperature, 72 h | 105/95 | |
| Silicon wafers | Oxidation: hydrogen peroxide, sulfuric acid, 10 min, 120 °C | Grafting: 2-[Methoxypoly(ethyleneoxy)-propyl]trimethoxysilane, toluene, room temperature, 18 h | 38 ± 2/34 ± 2 | [60] |
| Glass | Soaking in chromic acid overnight | Grafting: silanated PEG I/silanated PEG II, anhydrous toluene, 70 °C, overnight | I: 32.3 ± 2.9/18.5 ± 2.0; II: 49.3 ± 0.6/24.4 ± 2.5 | [61] |

**Table 1.** *Cont.*

| Substrate | Pretreatment | Reaction Conditions | $\theta_a/\theta_r$ (°) Water | Ref. |
|---|---|---|---|---|
| Silicon wafers | Hydrophilization: concentrated hydrogen chloride, hydrogen peroxide, deionized water, 80 °C, 15 min | PEG–silane coupling to silicon: hydrolysis of PEG–OSiCl$_3$, 120 min, room temperature | 42.33 ± 2.61 (CA) | [63] |
| | | | - | [64] |
| Si(100) surface | Cleaning: concentrated sulfuric acid, hydrogen peroxide; argon plasma | Grafting: PEGMA macromonomer, riboflavin, ethanol/water, UV illumination, 0.5–3.0 h | - | [65] |
| Glass sheets | Cleaning: detergent and water | Grafting: 2-[acetoxy (polyethyleneoxy) propyl]triethoxysilane (pH 5.5), TEOS, 10 min, 75 °C | 10.7 (CA) | [66] |
| Silicon wafers | Oxygen plasma | Grafting: DCDMS, toluene, room temperature, 1800 s | 104 ± 1/100 ± 1 | [67] |
| Silicon wafers | Reacting SiH$_4$ and O$_2$ gases in a PECVD reactor | PEG 400 (vapor), water plasma, 100 °C, vacuum | 25 ± 2 (CA) | [68] |
| Silicon wafers | Cleaning: oxygen plasma, 250 mTorr, 20 min | PDMS$^{2000}$, 100 °C, 24 h | 104/102 | [69] |
| Silicon wafers | Cleaning: UV/ozone | PEG$_{9-12}$-Si, TEOS, ethanol, aqueous HCl; spin-coated, dried at 80 °C for 3 h | 42 ± 2/35 ± 1 | [59] |

## 2.2. Surface Wettability Characterization

The contact angle is the most intuitive and accurate representation of the surface wettability and was measured by a contact angle measuring instrument (OCA 25, Dataphysics, Germany) in this work. As shown in Figure 2, the static contact angle (CA) of the plain-HI surface, the hydroxyl-HI surface, the plain-HO surface, and the ultra-SHI surface was 72° ± 3.6°, 37° ± 4.5°, 106° ± 5.2°, and 37° ± 2.3°, respectively. In addition to the static contact angle, the contact angle hysteresis (CAH), which is dependent on both the chemical uniformity and the physical smoothness, was measured in order to characterize the wettability of the solid surface. The contact angle hysteresis of a droplet on the surface was measured by changing the volume (V) of a water droplet in this work. The volume variation was performed as follows. A droplet with a volume of 5 µL was first pre-deposited onto the surface. Water to a volume of ΔV = 5 µL was injected into the droplet at a prescribed flow rate of 2 µL/s. The droplet was then left to rest for 30 s to allow enough time for the contact line to stabilize. The droplet volume was then increased by 5 µL each time until the volume reached 50 µL. Subsequently, water to a volume of ΔV = 5 µL was withdrawn from the droplet at the same flow rate and the above process was repeated until the droplet volume returned to 5 µL. The experiments were repeated at least five times for each surface.

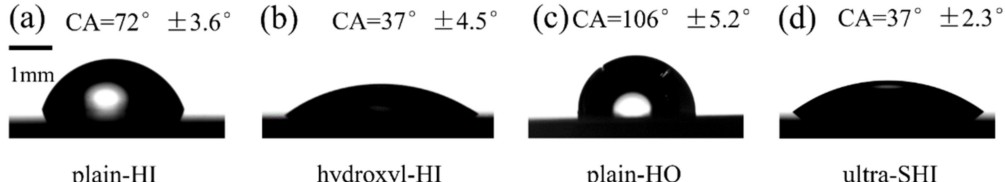

**Figure 2.** Surface wettability of the test surface. The static contact angle of a water droplet on the (**a**) plain-HI surface, (**b**) hydroxyl-HI surface, (**c**) plain-HO surface, and (**d**) ultra-SHI surface.

Figure 3 shows the variation in the base diameter (BD) and contact angle of the deionized water on each surface as a function of the droplet volume. The characteristics associated with the change in the base diameter are the forward and backward pinning, and the characteristics associated with the change in the contact angle are the advancing contact angle ($\theta_a$) and the receding contact angle ($\theta_r$). For the plain-HI surface (Figure 3a), the contact line of the growing droplet begins to move outward and the base diameter

increases with the increase in the droplet volume, but the static contact angle remains at $\theta_a = 81°$. When the droplet volume decreases, the contact line of the droplet is pinned and the static contact angle decreases to $\theta_r = 53°$. A further decrease in the droplet volume can cause the contact line to move inward while maintaining the receding contact angle, showing a large CAH of 28° for the plain-HI surface. For the plain-HO surface (Figure 3c), the static contact angle remains at $\theta_a = 112°$ and the contact line moves outward with the increase in the droplet volume. The contact line of the droplet is pinned and the static contact angle decreases to $\theta_r = 89°$ with the decrease in the droplet volume. Then, the static contact angle remains a receding contact angle and the base diameter decreases, showing a large CAH of 23° for the plain-HO surface. The hydroxyl-HI surface (Figure 3b) and the ultra-SHI surface (Figure 3d) have the same static contact angle but significantly different contact angle hysteresis values. For the hydroxyl-HI surface, with the increase in the droplet volume, the static contact angle of the droplet remains at $\theta_a = 37°$ but the base diameter increases. As the droplet volume decreases, the contact line is pinned with a decreasing contact angle until the volume decreases to 25 μL, which results in a CAH of 23°. For the ultra-SHI surface, the droplet continues to grow and the static contact angle remains at $\theta_a = 38°$ during the increase in the droplet volume. Similarly, the base diameter of the shrinking droplet decreases gradually and the contact angle remains at $\theta_r = 35°$, resulting in a CAH of 3°. Compared with the typical CAH, the droplet's base diameter does not have an invariant region. Due to the large CAH of the plain-HI, hydroxyl-HI, and plain-HO surfaces, contact line pinning will occur during the droplet shrinking process. The unique phenomenon of an ultra-low CAH was observed on the ultra-SHI surface without obvious contact line pinning, which provides a new surface wettability feature (low CAH for a hydrophilic substrate) for controlling the spreading behavior of impacting droplets.

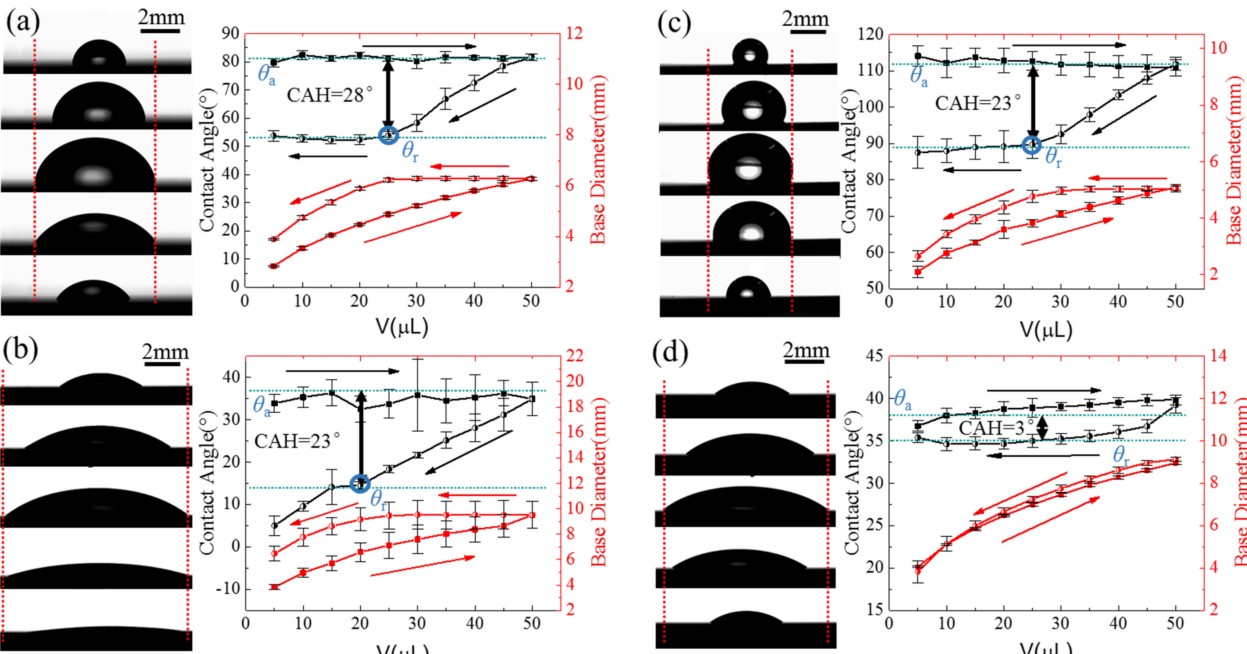

**Figure 3.** The variation in the base diameter (BD) and contact angle (CA) of a water droplet on the test surface as a function of the droplet volume. Advancing contact angle, receding contact angle, and contact angle hysteresis of a water droplet on the (**a**) plain hydrophilic surface, (**b**) hydroxylated hydrophilic surface, (**c**) plain hydrophobic surface, and (**d**) ultra-slippery hydrophilic surface.

### 2.3. Experimental System and Method

Figure 4 shows the experimental system for measuring the droplet impact and spreading characteristics on the substrate surface. The system is composed of droplet generation devices, object carrying devices, a high-speed camera (Memrecam HX-7S, NAC Image

Technology Inc., Tokyo, Japan), light sources, and a data acquisition device (Dell OptiPlex 3060, Xiamen, China). The needle is fixed on the high-precision linear slide and connected to the syringe through the liquid guide tube. The droplet falling frequency is controlled by the flow rate of the microinjection pump. The droplet impact velocity (Weber number) is controlled by the distance between the needle tip and the test surface blow. The droplet size is controlled by the needle diameter.

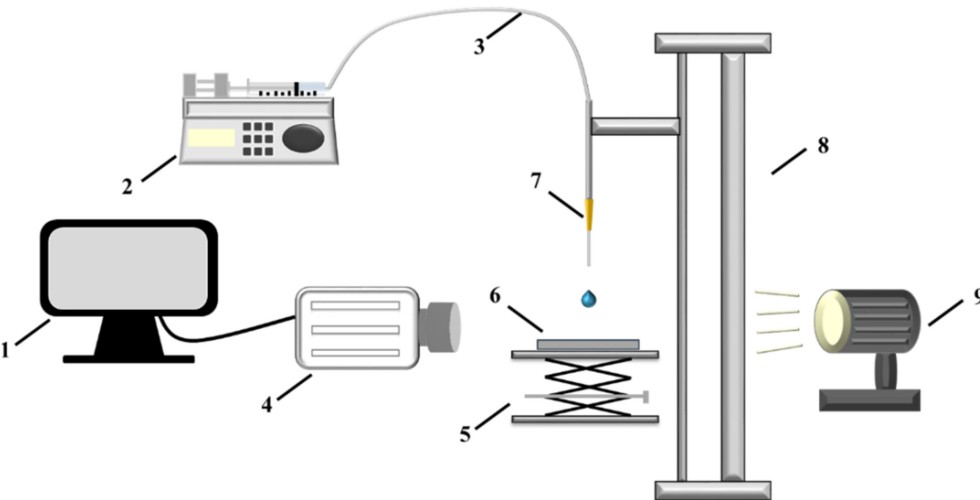

**Figure 4.** Experimental setup for measuring the droplet impact and spreading characteristics on the substrate surface: 1, Computer; 2, Injection pump; 3, Catheter; 4, High-speed camera; 5, Carrier table; 6, Surface; 7, Needle; 8, High-precision linear slide; 9, Light source.

The high-speed camera maintains the same horizontal height as the loading platform and takes images from the front. All experiments were conducted at room temperature (25 ± 0.5 °C), a relative humidity of 30% ± 5%, and atmospheric pressure. The droplet impact experiments were repeated three times to ensure the repeatability of the results. A high-speed camera was used in order to record the droplet impact and spreading dynamics at a frame rate of 5000 fps. The liquid was deionized water, and the density was $\rho_w = 1000$ kg·m$^{-3}$. The water–air surface tension was $\sigma = 72.8$ mN·m$^{-1}$. Pixel analysis was performed on the image of the droplet before the impact, and the droplet shape was found to be approximately elliptical during the falling process. The droplet equivalent diameter was calculated by $D_0 = (D_v D_h^2)^{1/3}$, where $D_0$ is the equivalent diameter of the droplet and $D_v$ and $D_h$ are the vertical and horizontal diameters of the droplet, respectively. The values of $D_v$ and $D_h$ were measured from the high-speed camera images using Image-Pro Plus. The pixel measurement error was 0.0248 mm. The initial diameter of the droplet before the impact was determined by averaging the results of one hundred experiments. The droplet equivalent diameter was 2.13 mm, and the relative error was 2.74%. Due to the influence of air resistance during the droplet falling process, the velocity of the droplet prior to the impact could be calculated from $U_0 = (g/\alpha \, (1 - \exp(-2\alpha h)))^{1/2}$, where $h$ is the falling height of the droplet and $\alpha = 3\rho_{air}C_f/(8\rho_{water}R)$. The air density was 1.205 kg/m$^3$, the Reynolds number was greater than 1000, and $C_f$ was regarded as a constant and equal to 0.44. The Weber number represents the ratio of the inertia effect to the interface tension effect and was defined as $We = \rho_w V_0^2 D_0/\sigma$, where $\rho_w$ is the density of the liquid, $\sigma$ is the liquid–gas surface tension, and $V_0$ and $D_0$ are the droplet velocity and diameter, respectively. The initial velocity of the droplet was 0.99–2.36 m/s, corresponding to a $We$ number range of 28.71–302.15. Because the droplet spreading process is very short in duration, the droplet evaporation was assumed to be negligible.

### 3. Results and Discussion

*3.1. Spreading Characteristics of the Droplet Impact on Horizontal Surfaces*

3.1.1. Effect of Surface Wettability on the Spreading Characteristics of the Droplet Impact

The spreading characteristics and evolution of the droplet impact on the horizontal surface with different wettability values are shown in Figure 5. The images captured by the high-speed camera were used to quantitatively analyze the droplet spreading diameter over time. The image of the last frame before the droplet impact on the surface was taken as the initial moment. After the droplet makes contact with the plain-HI surface (Figure 5a), the droplet begins to spread out, driven by the inertial force, resulting in the form of a lower lamella and upper wrinkles. The spreading diameter increases, and the center height decreases. The pressure of the droplet's edge bulge continuously increases, while the pressure of the droplet's middle part decreases. The impact pressure makes the droplet spread out and form a non-equilibrium liquid film [70]. The spreading diameter reaches its maximum value at 3.6 ms. At this time, the liquid film's thickness is not uniform, and the liquid film shows a thin layer in the center and a thicker edge near the contact line. Then, the radial retraction gradually forms a protrusion in the middle, driven by the surface tension, resulting in an increase in the peak of the central protrusion when it reaches the maximum recoiling height at 9.4 ms. After reaching the minimum spreading diameter in the retraction stage, the droplet repeats the processes of spreading and recoiling and finally reaches an equilibrium state, showing a spherical cap-like shape resting on the surface. The spreading behaviors on the hydroxyl-HI and plain-HO surfaces are similar to that on the plain-HI surface; however, subtle differences can be observed. The time required to reach the maximum spreading diameter is 6.2 ms and 3.6 ms for the hydroxyl-HI surface and the plain-HO surface, respectively. For the ultra-SHI surface, the time required to reach the maximum spreading diameter and the maximum recoiling height is 4.8 ms and 16.6 ms, respectively. The droplet on the plain-HO surface rapidly recoils and oscillates violently, reaching its maximum recoiling height at 13 ms. This is because the spreading diameter is smaller and more energy is used for the recoiling and oscillation processes due to the weak solid–liquid interaction force. Despite the fact that the contact angles of the droplet on the hydroxyl-HI surface and the ultra-SHI surface are similar, a large solid–liquid contact area is achieved more quickly on the ultra-SHI surface compared with the hydroxyl-HI surface. This is due to the lower contact angle hysteresis and the lower degree of dissipation during the spreading process. Additionally, it is easy for the impacting droplet on the ultra-SHI surface to transition from an advancing angle to a receding angle; thus, the recoiling and oscillation amplitudes are larger than those of the hydroxyl-HI surface.

To eliminate the influence of the initial droplet size, the dimensionless size ($\beta = D/D_0$, $\varepsilon = H/D_0$) was used to analyze the evolution of the droplet impact on the test surface. Figure 5e,f show the results for the spreading factor $\beta$ and the dimensionless recoiling height $\varepsilon$ at the *We* number of 28.71. After the droplet impacts the surface, the droplet spread is dominated by the inertial force. The spreading diameter and the droplet height show a trend of increasing and decreasing, respectively, and the droplet height curve on the surface fluctuates periodically with a decrease in the amplitude. In the initial spread regime, there is no significant difference between the droplet spreading curves and the recoiling height curves on the four test surfaces. The maximum spreading factor $\beta_{max}$ of the test surfaces exhibits an obvious difference. $\beta_{max}$ was found to be larger for the droplet on the surface with a smaller contact angle, causing a larger amount of energy dissipation during the spreading process. At the same time, the initial oscillation time increased from 4.4 ms to 7.6 ms, and the recoiling speed and the oscillation amplitude decreased when the contact angle decreased. It can be seen from Figure 5e that the $\beta_{max}$ of the ultra-SHI surface is the largest (2.84) when compared with that of the hydroxyl-HI, plain-HI, and the plain-HO surfaces (2.78, 2.29, and 2.09, respectively). When the droplets impact and make contact with the surface, the initial kinetic energy of the impacting droplet is not only used to overcome the viscous resistance but also transformed into the surface energy of the liquid film gradually due to the increase in the gas–liquid interface and liquid–solid interface. With

a decrease in the contact angle, a larger amount of interfacial energy and a larger amount of viscous dissipation occur during the spreading of droplets on hydrophilic surfaces. Compared with the hydroxyl-HI surface, although the contact angle and $\beta_{max}$ are similar, the time required for the droplet impacting upon the ultra-SHI surface to reach the $\beta_{max}$ is shorter, which is important for the rapid supply of liquid in many applications [38,71]. Thus, even for hydrophilic surfaces, low surface hysteresis can reduce the amount of dissipation caused by surface friction during the droplet spreading process and reach the maximum spreading area more quickly. After the impacting droplet reaches the maximum spread diameter, the contact angle changes from a dynamic advancing contact angle to a receding contact angle, which causes a delay in the onset of the droplet's retraction. Then, the spreading factor starts to decrease and finally remains at a constant value. A slight oscillation process was observed on the droplet on the ultra-SHI surface during the retraction process, while almost no oscillation was present on the hydroxyl-HI surface.

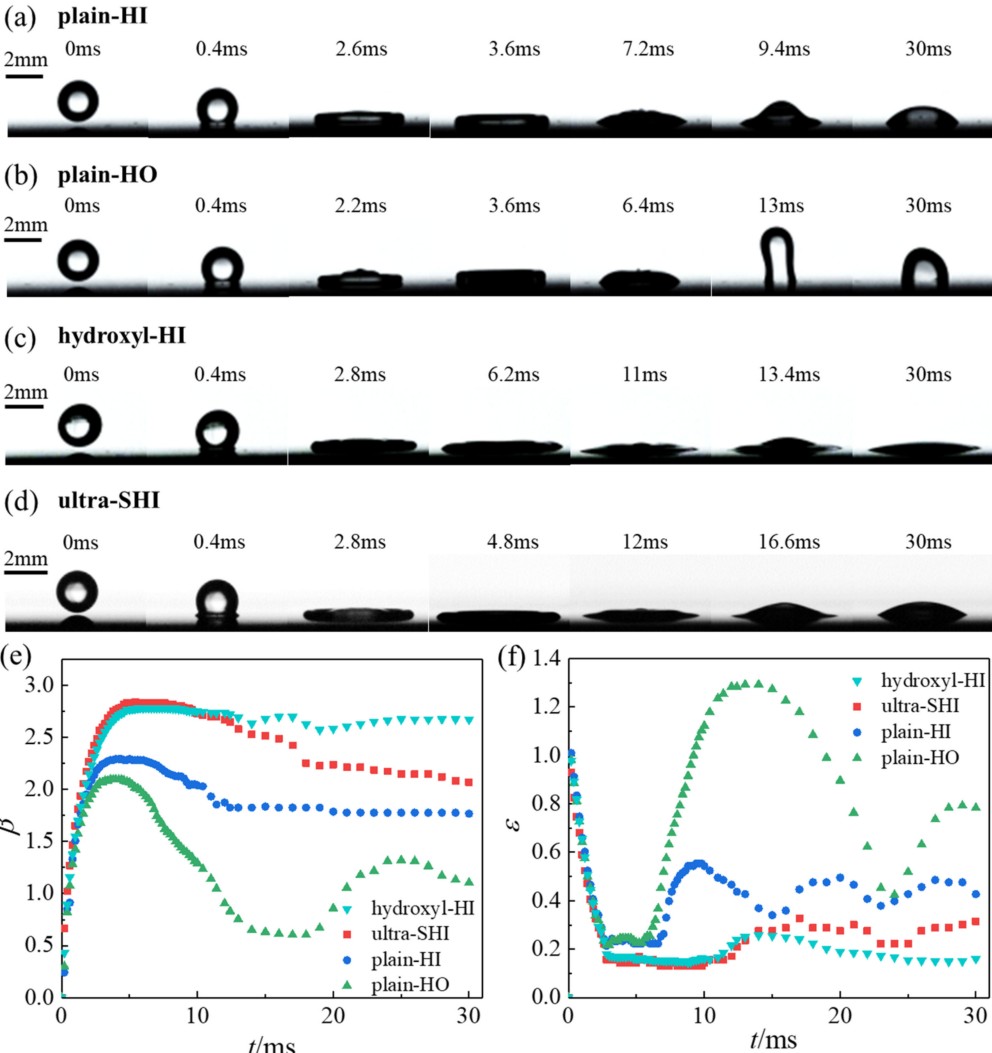

**Figure 5.** Spreading characteristics of the droplet on the horizontal surface with different static contact angles and contact angle hysteresis values. Spreading dynamics of a droplet impacting upon the (**a**) plain-HI surface, (**b**) plain-HO surface, (**c**) hydroxyl-HI surface, and (**d**) ultra-SHI surface. *We* = 28.71. Comparison of (**e**) the spreading factor $\beta$ and (**f**) the dimensionless recoiling height $\varepsilon$ of the four test surfaces.

### 3.1.2. Effect of We Number on the Spreading Characteristics of the Droplet Impact

Figure 6a shows the maximum spreading factor $\beta_{max}$ and the maximum recoiling height $\varepsilon_{max}$ as a function of the *We* number for the test surfaces. With a decrease in the contact angle, a larger $\beta_{max}$ and a smaller $\varepsilon_{max}$ were observed because less energy was used to drive the droplet retraction process. With a contact angle similar to that of the hydroxyl-HI surface, the recoiling speed is faster and the $\varepsilon_{max}$ is larger for the droplet on the ultra-SHI surface, which reduces the resistance time for more energy to drive the recoiling process with lower contact angle hysteresis. For the three hydrophilic surfaces (the plain HI, hydroxyl-HI, and ultra-SHI surfaces), the maximum recoiling height $\varepsilon_{max}$ decreases with the increase in the *We* number. For the hydrophobic surface (the plain-HO surface), however, the maximum retraction height increases with the increase in the *We* number because more surface energy is available for the oscillation process. Figure 6b shows the inertia time required to reach the maximum spreading diameter $t_{Dm}$ and the time interval during which the maximum spreading diameter $t_{Dm}-t_{0.99Dm}$ was maintained (defined as the time interval during which the droplet remained at $\beta > 0.99\beta_{max}$ after reaching $\beta_{max}$) as a function of the *We* number on the test surfaces. With the increase in the *We* number and the contact angle, both the $t_{Dm}$ and the $t_{Dm}-t_{0.99Dm}$ of the test surfaces decrease. However, they are greatly affected by the contact angle hysteresis when the contact angles are similar. For the hydroxyl-HI surface and the ultra-SHI surface with similar contact angles but different CAH values, the $t_{Dm}$ and $t_{Dm}-t_{0.99Dm}$ are different. Specifically, for the ultra-SHI surface with a smaller CAH value, the time interval during which the maximum spreading diameter was maintained is reduced while a similar $\beta_{max}$ is reached more quickly with a smaller $t_{Dm}$ compared with the hydroxyl-HI surface. In addition, the time required for the dynamic contact angle to change from an advancing contact angle to a receding contact angle is shorter, which means that the recoiling process occurs earlier.

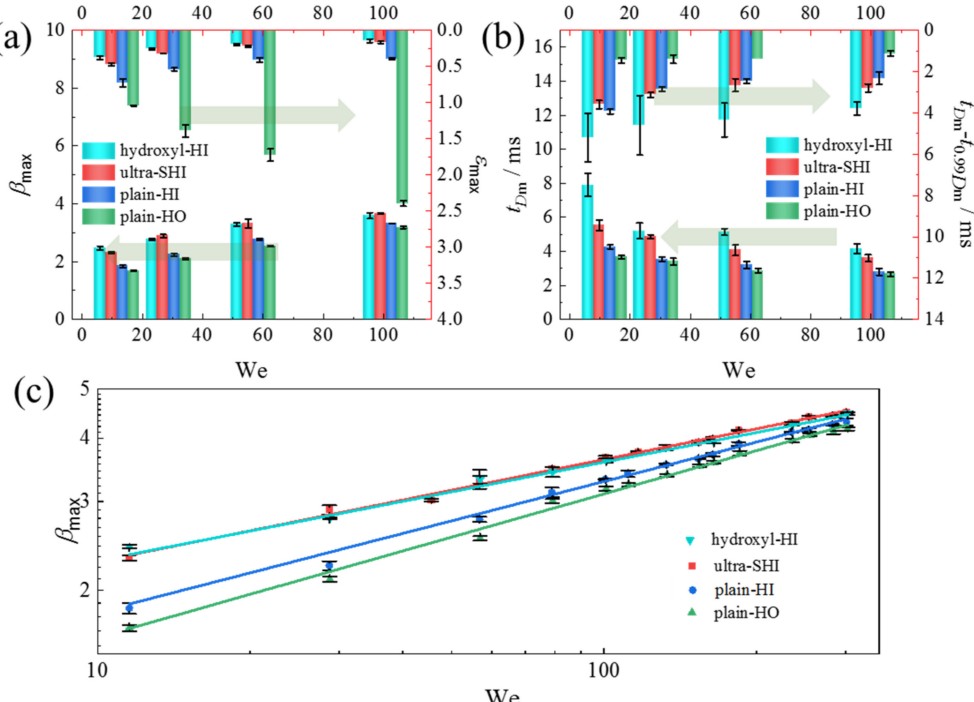

**Figure 6.** Effect of the *We* number on the spreading characteristics of the droplet impact. (**a**) Maximum spreading factor $\beta_{max}$ and maximum dimensionless recoiling height $\varepsilon_{max}$ as a function of the *We* number on the test surfaces. (**b**) Inertia time required to reach the maximum spreading diameter $t_{Dm}$ and the time interval during which the maximum spreading diameter $t_{Dm}-t_{0.99Dm}$ was maintained as a function of the *We* number on the test surfaces. (**c**) Variation in the maximum spreading factor $\beta_{max}$ as a function of the *We* number on the test surfaces.

According to the energy conservation of the droplet spreading process, it was assumed that the kinetic energy of the impacting droplet would be completely transformed into surface energy without energy dissipation, and the maximum spreading diameter would follow the one-half power law $D_{max}/D_0 \sim We^{1/2}$. For a low-viscosity liquid, the prediction model of the maximum spreading diameter can be modified to $D_{max}/D_0 \sim We^{1/4}$ [72]. In addition, many prediction models have been proposed to calculate the maximum spreading diameter under different conditions [73,74]. It has been found that the inertial, capillary, and viscous forces are important in most practical cases, and the droplet spreading characteristics depend on the relative importance of each force. Equating inertial forces with capillary or viscous forces alone cannot accurately predict the droplet spreading characteristics [74,75]. The impact factor of $WeRe^{-4/5}$ has been also proposed to determine the linear relationship. When $WeRe^{-4/5} > 1$, $D_{max}/D_0$ is only related to the Reynolds number, namely $D_{max}/D_0 \sim Re^{1/5}$ in the viscous regime. When $WeRe^{-4/5} < 1$, the Reynolds number can be omitted and $D_{max}/D_0$ corresponds to the $We^{1/4}$ scaling for a low-viscosity liquid. Figure 6c compares the maximum spreading factor of the droplet impacting upon the test surface in the range of *We* numbers between 10 and 300. In all cases, the maximum spreading diameter increases with the increase in the *We* number. For the droplet impact on the plain-HI and plain-HO surfaces, $D_{max}/D_0$ conforms to the $D_{max}/D_0 \sim We^{1/4}$ mentioned in the literature. When the impact factor satisfies $WeRe^{-4/5} < 1$, $D_{max}/D_0$ conforms to $We^{1/5}$ for a droplet impacting upon the hydroxyl-HI and ultra-SHI surfaces.

## *3.2. Spreading Characteristics of the Droplet Impact on Inclined Surfaces*

### 3.2.1. Effect of Surface Wettability on the Spreading Characteristics of the Droplet Impact

As shown in Figure 7a, a droplet impacting upon a solid surface usually has different angles with the substrate in many practical applications, such as spray cooling. An analysis of impacting droplets on inclined surfaces was performed to investigate the spreading characteristics of impacting droplets, and the results were compared with those on horizontal surfaces. To quantitatively describe the droplet impacting upon the inclined surface, a coordinate system was defined and is shown in Figure 7b. The point where the droplet first impacts the surface was defined as the origin, and the positive direction of the *x*-axis is parallel to the downward direction of the inclined surface. In the process of the droplet's movement, the most anterior point and the last point on the surface were defined as the foremost point and the rearmost point, respectively. The distance between the foremost point and the impact point was defined as $x_{fo}$, and the distance between the rearmost point and the impact point was defined as $x_{re}$. The contact diameter between the droplet and the substrate during the droplet's movement was defined as $D_s$. The distance from the droplet impact point to the center point of the moving droplet was defined as the gliding distance *L*.

When a droplet impacts a horizontal surface, the droplet spreads and contracts symmetrically without the movement of the droplet's center position. In comparison, when a droplet impacts an inclined surface, the droplet spreads asymmetrically and glides along the inclined substrate. Overall, the droplet impact on inclined hydrophilic and hydrophobic surfaces goes through five typical regimes: a kinetic regime, a spreading regime, a gliding regime, a retraction regime, and a stable wetting regime. According to previous studies [76,77], the duration of the kinetic regime can be ignored. As shown in Figure 7c–e, the spreading behavior of the droplet impacting upon the test surface in the spreading regime after impacting upon the inclined surface is similar to that of the flat surface. The droplet immediately spreads out in the radial direction after impacting upon the surface inclined at 45°, and a thin lamella is formed at the bottom. At this time, the initial kinetic energy of the impacting droplet begins to transform into surface energy and viscous dissipation. The droplet's morphology does not show obvious asymmetry. The high-pressure region changes from being in the inner central region to being at the foremost and rearmost points of the droplet during the spreading regime. The velocity component in the *x*-axis decreases significantly due to internal shear stress and wall friction. Because the pressure on the

circular edge of the liquid is higher than that on the inside of the concave, the velocity of the liquid is also reduced by the surface tension [78]. The spreading diameter keeps increasing, resulting in a pancake shape. When the spreading regime is completed (at $t = 4$ ms), the droplet begins to glide downward. Both the foremost and rearmost points of the droplet glide but at different speeds due to the action of tangential gravity and kinetic energy. The foremost point glides faster than the rearmost point, and the contact diameter of the droplet further increases. The droplet spreads asymmetrically across the surface, forming a bulge at the leading edge of the droplet and leaving a thin trail behind it. For the hydrophilic surface with high surface energy, e.g., the plain-HI surface (Figure 7c) and the ultra-SHI surface (Figure 7e), the rearmost point of the impacting droplet is fixed on the surface. The kinetic energy associated with the tangential component of the impact velocity drives the foremost point to glide down along the inclination angle. When the velocity of the foremost point reaches zero, the gliding regime ends. After that, the droplet begins to retract, the rearmost point rapidly retracting driven by surface tension but the foremost point remaining stationary. On the other hand, the rearmost point of the droplet is not fixed on the plain-HO surface with low surface energy, showing the phenomenon of retracting in the gliding regime (Figure 7d). Before the final stability value is reached, the impact energy is dissipated by the retraction motion, and, finally, the droplet reaches an equilibrium state and enters the stable wetting regime with pinning on the surface. Due to the smallest contact angle hysteresis of the droplet on the ultra-SHI surface, the impacting droplets continue to slip after the retraction regime ends.

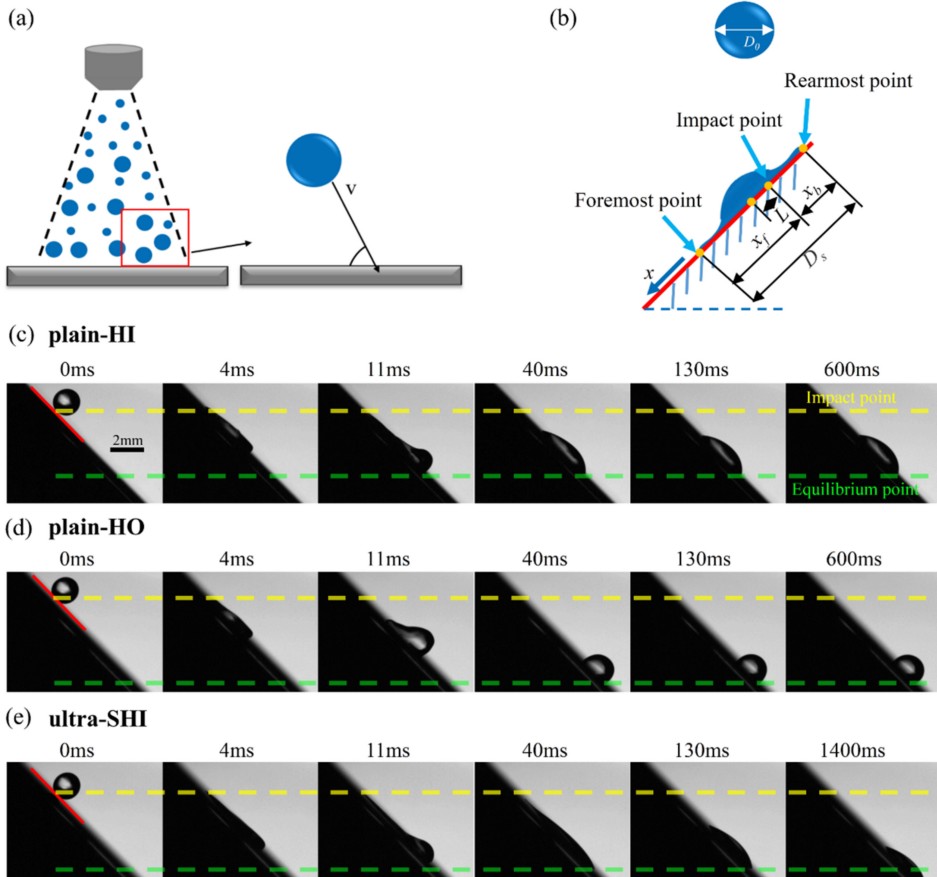

**Figure 7.** Droplet spreading and slipping dynamics on inclined surfaces with different surface wettability values. (**a**) Schematic of the droplet impact on the surface at different angles. (**b**) Analysis of the droplet impact on an inclined surface. Spreading dynamics of the droplet impact on inclined surfaces, including the (**c**) plain-HI surface, (**d**) plain-HO surface, and (**e**) ultra-SHI surface. *We* = 28.71. The inclination angle of the surface is 45°.

Figure 8a shows the displacements of the foremost point $x_{fo}$ and the rearmost point $x_{re}$ of the droplet over time $t$ on the test surface with different surface wettability values. The displacements of the foremost point and the rearmost point of the droplet are asymmetric to the impact point. When the inclination angle is 45° and $We = 28.71$, for the plain-HO and plain-HI surfaces, the displacement of the foremost point $x_{fo}$ increases continuously over time until it reaches the maximum value and then stabilizes. The displacement of the rearmost point $x_{re}$ firstly decreases to a minimum value, then slowly increases to a positive value, and finally reaches a stable value. This is because the rearmost point of the droplet first spreads out in a direction above the impact point, then moves down in the retraction regime, and eventually falls below the impact point. The spreading curves of the droplet impacting upon the inclined ultra-SHI surface are similar to those of the droplet impacting upon other surfaces in the spreading, gliding, and retraction regimes. In the retraction regime, $x_{fo}$ remains unchanged and then $x_{re}$ increases continuously. However, $x_{fo}$ and $x_{re}$ increase at the same time after the retraction regime ends, demonstrating slipping within a stable wetted area. In the spreading regime, all the droplet spreading curves overlap with each other, and the influence of wettability on spreading is not obvious. The rearmost point of the droplet no longer recedes but glides downward along the inclined surface when the droplet enters the gliding regime. A difference in the velocity between the rearmost and foremost points was observed due to the different surface wettability values. As shown in Figure 8a, the displacement of the foremost point for the droplet impacting upon the plain-HO surface is larger than that for the hydrophilic surfaces due to the weaker solid–liquid interaction. The smaller the contact angle of the solid surface is, the larger the displacement of the foremost point is. For the rearmost point, greater spreading along the negative direction, namely $|x_{re}|$, was found with the decrease in the contact angle. A decrease in the static contact angle leads to the extension of the spread duration and delay in the initiation of the retraction regime. The equilibrium state is reached more slowly. Due to the large hysteresis of the droplet on the plain-HI and plain-HO surfaces, the impacting droplet finally reaches a static state on the surface and the displacement of the foremost and rearmost points remains unchanged. While the contact angle hysteresis of the ultra-SHI surface is small, the droplet moves downward because the gravity component on the $x$-axis is large enough to overcome the frictional resistance of the droplet gliding along the surface. Even after the retraction regime ends, the displacement of the foremost and rearmost points continues to increase.

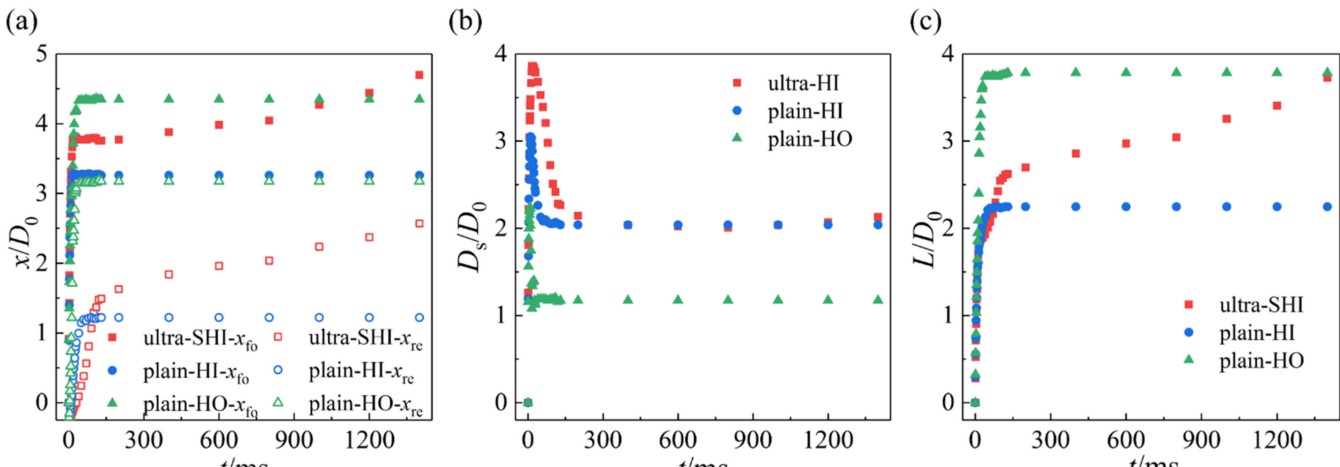

**Figure 8.** Evolution and slipping distance of the droplet impact on inclined surfaces. (**a**) Displacement of the foremost point $x_{fo}$ and the rearmost point $x_{re}$ of the droplet over time $t$ on the test surface. (**b**) Spreading diameter of the droplet $D_s/D_0$ over time $t$ on the test surface. (**c**) Slipping distance of the droplet $L/D_0$ over time $t$ on the test surface. $We = 28.71$. The inclination angle of the surface is 45°.

Figure 8b shows the spreading diameter of the droplet $D_s/D_0$ over time $t$ on the test surface. The maximum contact diameter of the droplet ($D_{smax}$) was defined as the contact diameter corresponding to the peak value in the curve. There is no significant difference between the three curves in the initial spread regime. The collision law of the droplet is the same as that of the horizontal surface: the $D_{smax}$ and the amount of time the droplet takes to reach $D_{smax}$ are larger when the contact angle is decreased. After reaching $D_{smax}$, the droplet retracts and the spread diameter begins to decrease, driven by surface tension. The retraction speed slows down and the time required to reach stability is longer with the decreasing contact angle. Figure 8c shows the gliding distance of the droplet $L/D_0$ over time $t$ on the test surface. The decrease in the contact angle results in a longer wetted length of the impacting droplet along the $x$-axis and a decreased gliding distance. In this case, the downward movement speed of the droplet on the plain-HO surface is slightly faster, and, finally, the surface is stably wetted and the gliding distance does not change. Different from the droplet-pinning phenomena on the plain-HI and plain-HO surfaces, the droplets impacting upon the ultra-SHI surface can continue to move downward and the gliding distance continues to increase after the retracting regime ends.

3.2.2. Effect of Inclination Angle and We Number on the Spreading Characteristics of the Droplet Impact

Figure 9a shows the dimensionless displacement of the foremost point $x_{fo}$ and the rearmost point $x_{re}$ of the droplet over time $t$ on the ultra-SHI surface with different inclination angles at $We$ = 28.71. It can be seen that $x_{fo}$ increases with the increase in the inclination angle, and $x_{re}$ in the negative direction decreases with the increase in the inclination angle. For example, when the tilt angle changes from 30° to 45°, $x_{fo}$ increases from 2.10 to 3.19 while $|x_{re}|$ decreases from 0.75 to 0.21 at $t$ = 9 ms. The foremost point and the rearmost point with a smaller tilt angle reach a stable value faster than the droplet on the surface with a larger tilt angle. The displacement of the rearmost point on the surface with a larger tilt angle reaches a positive value faster than that when the tilt angle is small. As the $We$ number increases, $x_{fo-max}$ increases for the surfaces with a tilt angle of 45° and 30°, which is because the initial velocity component of the impacting droplet along the slope direction increases. The maximum displacement of the rearmost point $|x_{re-max}|$ along the negative x-axis does not change with the increase in $We$ for the surface with the angle of 45°, while $|x_{re-max}|$ increases with the increase in $We$ for the surface with the angle of 30°, as shown in Figure 9b. This is because the components of the droplet impact velocity and the gravity in the direction of inclination promote the movement of the foremost point but inhibit the spreading of the rearmost point. The droplet gliding along the surface with the larger tilt angle has greater momentum. Compared with the spreading of the droplet along the rearmost point, the gliding is more obvious and faster, which causes the rearmost point of the droplet to rapidly exceed the height of the initial impact point. The droplet starts to retract in the gliding regime, resulting in an overlap between the two regimes and an increased gliding duration. In addition, the droplet enters the stable wetting state later.

Figure 9c shows the variation in the maximum spreading diameter $D_{smax}$ as a function of the $We$ number for the droplet impact on the test surface with different inclination angles. For the ultra-SHI surface, $D_{smax}$ increases with the increase in the inclination angle of the surface and the $We$ number of the impacting droplet. This is because the motion of the droplet in the spreading regime is mainly determined by the inertial force. The increase in the $We$ number leads to an increase in the tangential inertial force component, increasing $D_{smax}$. $D_{smax}/D_0$ was fitted with the $We$ number for the surface with different tilt angles, and $D_{smax}/D_0 \sim We^{1/4}$ was obtained. As can be seen from Figure 9b, the gliding distance exhibits an upward trend with the increase in the tilt angle of the surface and the $We$ number of the droplet. From collision to resting on the surface, the initial kinetic energy of the droplet is dissipated by internal viscosity and friction with the surface. The initial kinetic energy and gravity components of the droplet are larger when the tilt angle and $We$ number increase. Therefore, the droplet needs to glide a longer distance to dissipate the

initial kinetic energy and reach a static state. For the ultra-SHI surface, the slip distance of the droplet continues to increase after the retracting regime ends. As shown in Figure 9d, $L/D_0$ shows a linear relationship with time; that is, the droplet slips uniformly under different *We* numbers.

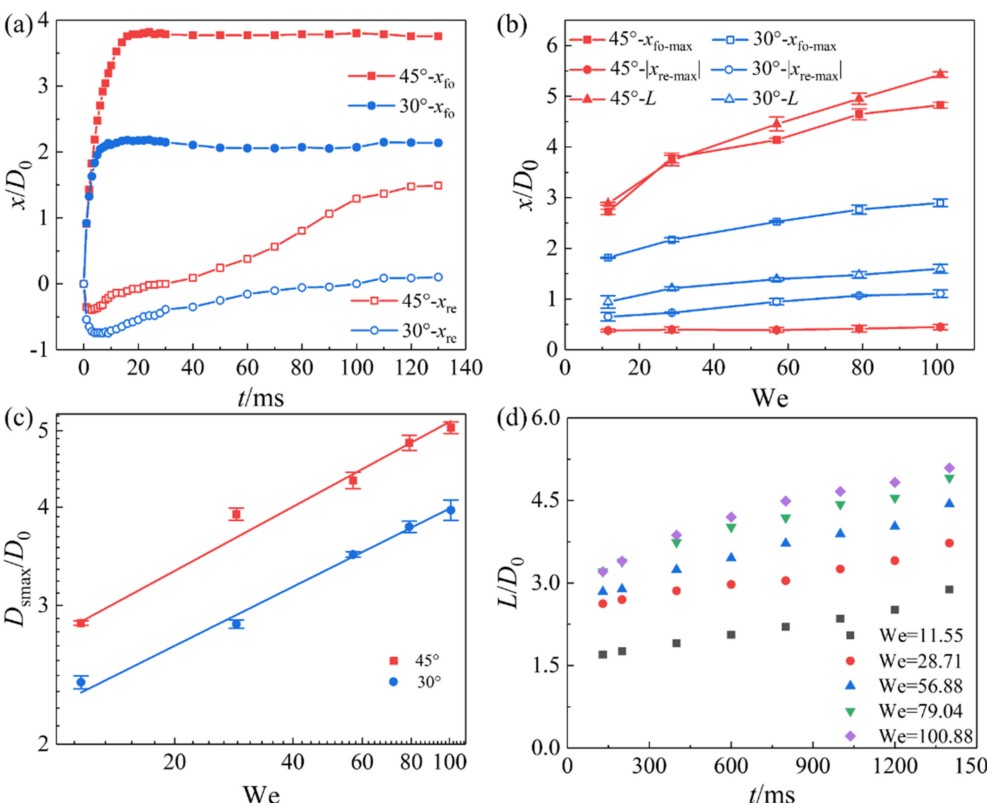

**Figure 9.** Effect of inclination angle and *We* number on the spreading and slipping of droplets on the ultra-slippery hydrophilic surface. (**a**) Displacement of the foremost point $x_{fo}$ and the rearmost point $x_{re}$ of the droplet over time *t* on the test surface with different inclination angles. (**b**) Effect of the *We* number on the maximum displacement of the foremost point $x_{fo\text{-}max}$ and the rearmost point $|x_{re\text{-}max}|$ of the droplet on the test surface with different inclination angles. (**c**) Variation in the maximum spreading diameter $D_{smax}$ as a function of the *We* number for the droplet impact on the test surface with different inclination angles. (**d**) Dimensionless slipping distance $L/D_0$ over time *t* with different *We* numbers on the surfaces.

## 4. Conclusions

In this work, we demonstrated the spreading of a droplet impacting without contact line pinning upon an ultra-slippery hydrophilic solid surface having a static contact angle of 37° and a small contact angle hysteresis of 3°. The results reveal that the maximum spreading factor of impacting droplets is mainly dependent on the static contact angle. The maximum spreading factor increases with the decrease in the static contact angle. For the hydrophilic surface with a similar static contact angle, the decrease in contact angle hysteresis helps the impacting droplet reach the maximum spreading diameter faster in the inertial spreading regime and decreases the time interval during which the maximum spreading diameter is maintained. Compared with the dependence of the spreading diameter and sliding distance on the static contact angle, the contact angle hysteresis affects the motion state of a droplet impacting upon an inclined surface after the retracting regime ends. Different from the droplet-pinning phenomena on the plain hydrophilic and plain hydrophobic surfaces, the droplets impacting upon the ultra-slippery hydrophilic surface can continue to slip at a constant speed on the inclined substrate. For the droplet impacting upon the ultra-slippery hydrophilic surface, the maximum contact

diameter of the impacting droplet conforms to $D_{\max}/D_0 \sim We^{1/5}$ on the horizontal substrate and to $D_{\mathrm{smax}}/D_0 \sim We^{1/4}$ on the inclined substrate. The outcomes of this work not only demonstrate the important effect of the contact angle and contact angle hysteresis on droplet spreading characteristics but also represent a pathway to advanced surfaces and coatings for controlling dynamic interactions of the droplet impact.

**Author Contributions:** Conceptualization, R.W. and X.M.; methodology, Y.S. and Z.Y.; validation, Y.Y., Q.W. and J.C.; investigation, Y.S., Y.Y. and S.W.; resources, X.M.; data curation, Y.S. and Q.W.; writing—original draft preparation, Y.S.; writing—review and editing, R.W. and X.M.; visualization, Y.S., Z.Y. and Q.W.; supervision, R.W. and X.M.; funding acquisition, R.W. and X.M. All authors have read and agreed to the published version of the manuscript.

**Funding:** This research was funded by the National Natural Science Foundation of China (No. 52006025 and No. 51836002) and the Fundamental Research Funds for the Central Universities (No. DUT20RC(3)016).

**Institutional Review Board Statement:** Not applicable.

**Informed Consent Statement:** Not applicable.

**Data Availability Statement:** The raw/processed data required to reproduce these findings cannot be shared at this time due to legal or ethical reasons.

**Conflicts of Interest:** The authors declare no conflict of interest.

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
