# Peer review of "Droplet Spreading Characteristics on Ultra-Slippery Solid Hydrophilic Surfaces with Ultra-Low Contact Angle Hysteresis"

_coatings, doi:10.3390/coatings12060755_

Round 1

Reviewer 1 Report

The manuscript entitled "Droplet Spreading Characteristics on Ultra-slippery Solid Hydrophilic Surfaces" was reviewed. This work and as-obtained results are interesting. This work reports rapid spreading of droplet impacting without contact line pinning on an ultra-slippery hydrophilic solid surface having an equilibrium contact angle of 37° and low contact angle hysteresis of 3°. The spreading characteristics are systematically compared with that on the plain hydrophilic, hydroxylated hydrophilic, and plain hydrophobic surfaces. I have the following comments;

  1. The title is informative and relevant, it could be more specific.
  2. The Abstract part is weak and it must be more informative by including more mathematical findings and more powerful explanations.
  3. The idea of the research seems to be interesting but the set goals are not achieved. The authors should clarify in the Introduction section, what is the novelty of their work?
  4. What the main significance of paper in comparison is of relates published works?
  5. Introduction writing part is not satisfactory. Need to be improved.
  6. I have read and evaluated the manuscript and in my opinion the submission does not yet sufficiently justify publication. In order to fix this problem, addition of description on recent development in the field of research topic with citing recent comprehensive papers would be important. Discuss the shortcomings of previous work and the gaps and how this work intends to fill those gaps. Related references should be cited:

Inorganic Chemistry Communications 9:2 (2006), 175-179; Carbohydrate Polymers 229 (2020) 115428;Materials science and engineering: C 76 (2017) 1085-1093; Ceramics International 46:11 (2020) 17186-17196;Process Safety and Environmental Protection 93 (2015) 282-292; Composites Part B: Engineering 174 (2019) 106930; Ultrasonics sonochemistry 42 (2018) 171-182; Journal of alloys and compounds 496 (2010), 638-643

  1. In “Result and Discussions” section, authors should add enough references to support results of their works.
  2. The figures in whole manuscript have poor qualities, leading to difficulty in reading and understanding the manuscript. Also, the font size, style etc., should be consistent. Please refine.
  3. The authors explained irrelevant and unnecessary subjects. There is no scientific clarification and they failed to provide scientific explanation.

Manuscript has however some grammatical mistakes which need to be improved. I recommend publication of this article after minor revisions and would like to see the revised version of paper before publication.

Author Response

Please see the attached response letter.

Reviewer 2 Report

The manuscript entitled "Droplet Spreading Characteristics on Ultra-slippery Solid Hy- 2 drophilic Surfaces". I do see a good importance in this work; the topic is relevant for material and surface fields. However, when is compared to the previous reports published in literature, this work lacks some important information and discussion. Before I recommend its acceptance, few points must be clarified and a moderate revision is needed.

Some other issues that need to be addressed are:

  1. The authors satisfactorily explained the problem that they wanted to solve and the contributions of the study in the abstract.
  2. The main problem statement and justification for the research have been clearly stated. Good job. However, I feel that is not clear the contribution of the manuscript to the empirical literature. Could you be clearer?
  3. The authors should mention on the concept of this work with the progress against the most recent state-of-the-art similar studies.
  4. The limitation of this study needs to be provided as well.
  5. How was the quality control of the measurements obtained by the authors?
  6. Limitations in the suggested approach should be discussed in the conclusions section. The conclusion section appears to be just a detailed summary of results/observations. All conclusions must be convincing statements on what was found to be novel, impactful based on the strong support of the data/results/discussion.

Author Response

(The authors gave the same response as above.)

Reviewer 3 Report

Below you will find my impressions concerning the manuscript Droplet Spreading Characteristics on Ultra-slippery Solid Hydrophilic Surfaces from Rongfu Wen.

I am convinced that the topic is extremely interesting for the readers of the journal and that the experimental work was carried out with great care (especially the high speed dynamic wetting behaviour investigations). Unfortunately, I am missing some information, some points are unclear and a comparison with statistically more robust studies conducted in the last 10 years on very similar systems has been omitted. Without such clarification, I cannot support publication at this time. Due to lack of time and the large number I cannot address all necessary points but I can provide you with a few guidelines / suggestions to improve the work.

Revisions

1.     Abstract/Introduction have to be corrected which contains a number of vague statements which intended meaning cannot be understand without considering the whole manuscript (some of them are simply confusing and redundant)
o   Static contact angle is much less confusing then equilibrium contact angle.
o   plain hydrophilic, hydroxylated hydrophilic, and plain hydrophobic surfaces. -- ??!! native silicon wafer (SiO2 surface), plasma treated silicon wafer (activated SiO2 surface), mono-perfluorooctylsiloxane (FOS) silcon wafer ,… (your cleaning/pretreatment might be considered not optimal by some readers)
o   spreading normally is CA 0 deg (not optically masurable). Hence use quotation marks
o   Did you use a lubricant or not. You might consider to streamline the introduction a bit, more focus on the system / surface you investiagted
2.     Layout
o   “Materials/Chemicals” section is missing
o   Use subsection in “Surface Fabrication and Methods” and keep an eye on singular and plural you fabricated more than one surface ! (e.g. “surface pre-treatments and modifications”)
3.     Missing references
o   Silanes to siloxanes formation, mechanism and so on.
o   Even within the last 10 years a number publications of mostly basing on HPDSA used and analysed in a statistical manner “moving” droplets on silicon wafers (native, modified even chemical surface patterns on the dynamic wetting behaviour on flat and silanized silicon wafers) even published in this journal
o   Wetting PROPERTIES OF SILICON SURFACES, Hermansson (pretreatment and SiO2 is a long-known issue)
4.     Missing infromation
o   For the «quasistatic experiments » using the OCA25, software, camera procedure (ellipse fitting, laplace), any modifications to the measuring chamber, or open to atmosphere, temperature, vapor control, pretreatment (everything is necessary if you indicate a thermodynamic equilibrium CA)…  it is shown in at least one publication the procedures include in SCA25 lacks in precision and robustness of more sophisticated procedures like HPDSA
o   Figure 3 How was the volume increased, decreased, rate, video analysis,… ?!
o   Page 3 not clear .... ?? Where does the water came from ?? Dynamic ... this clearly isn't dynamic analysis the authors should take a look at the works of Heib and Schmitt
o   178 if something moves what is the velocity?
o   Line 194 reasoning is not clear “due to the large CAH
o  184,185 confusing, the do not look similar
o   how many experiments were performed?
o   Difference in Figure 2 and 3 what is your “equilibrium CA” looks like some apparent CA advancing angle but the figure 3 suggest otherwise
o   What is the average roughness values Ra and Rq of the (coated) silicon wafers, the thickness of the oxide layers (e.g. procedure can be found in mentioned missing references.)?
o   Any statistical procedures, repetition experiments (native and activated silicon wafers will results in analyzable differences in their wetting-behavior in dependence on the history of the material).
o   Experimental procedure is incomplete and not acceptable (oxide layers are not explained investigated or even a reference is named). You will find related information, detailed and carefully analyzed wafer/microchips in the publications of Munief + wafer in WWW. You did not even mention the area of the wafer, the volume of the chamber, and so on
o   Repeat measurements and robustness of the dynamic Measurements (drop impact) seems not to be investigated. The text speaks about only one experiment (“After the droplet impacts”) not about a distribution of droplets with similar comparable behavior.
o   What about the error of the dosing volume? (Developer of printing heats would be not so happy if it was so simple). The OCA25 should be able to monitor the drops and data analysis is quite simple.
o   You could think about analysis of the curvature distribution of the meniscus instead of using “transform into surface energy”

Author Response

(The authors gave the same response as above.)

Round 2

Reviewer 1 Report

I will consider publishing your paper entitled "Droplet Spreading Characteristics on Ultra-slippery Solid Hydrophilic Surfaces with Ultra-low Contact Angle Hysteresis"; with inclination angle and We increase, the droplet gliding distance becomes longer. The reduction of the contact angle hysteresis allows droplets to slip at a uniform speed on the hydrophilic surface rather than pinning to the surface. The findings of this work not only show the important role of surface wettability on droplet spreading characteristics but also present a pathway for controlling dynamic interactions of impacting droplets with ultra-slippery hydrophilic surfaces. After reading authors’ responses, I feel that my initial concerns have been partially addressed, but there are still some minor concepts I cannot agree with.

  1. In its current state, the level of English throughout the manuscript needs language polishing. Please check the manuscript and refine the language carefully.
  2. Make sure all abbreviations are written out in full the first time used. This is particularly important in the abstract and in the conclusions, but work through the entire ms carefully from this perspective.
  3. Each section (Introduction, Experimental, Discussion and Result, Conclusions) should be more specific and detailed.
  4. The structure of the manuscript might need a minor adjustment for a better understanding.
  5. In the “Introduction” section, general description on the importance of manuscript topic is poor. Therefore, the importance of this work cannot be well recognized from general readers. The introduction can be improved by providing a more critical discussion of recent related literature. Related references should be cited:

- Journal of Colloid and Interface Science, 497 (2017) 57-65. - Materials, 10 (2017) 697.- Cellulose, 27 (2020) 9559–9575.- Cellulose, 27 (2020) 4691–4705. Ultrasonics sonochemistry 39 (2017) 494-503; Journal of Industrial and Engineering Chemistry 21 (2015) 1301-1305

  1. The quality and resolution of figures are not suitable.
  2. The Conclusions section is too long. It should be kept short and must be fully supported by the results reported.

Reviewer 3 Report

The type, quality and origin of the materials (chemicals) are still open and a comparison with more recent literature has not been made. 

It must be clearly stated that the original surface is a native oxide layer. 
